# LEAVE NO TRACE: LEARNING TO RESET FOR SAFE AND AUTONOMOUS REINFORCEMENT LEARNING

**Benjamin Eysenbach**[∗ †]**, Shixiang Gu**[† ‡ ††]**, Julian Ibarz**[†]**, Sergey Levine**[† ‡‡]
[†]Google Brain
[‡]University of Cambridge
[††]Max Planck Institute for Intelligent Systems
[‡‡]UC Berkeley
{eysenbach,shanegu,julianibarz,slevine}@google.com

## ABSTRACT

Deep reinforcement learning algorithms can learn complex behavioral skills, but real-world application of these methods requires a large amount of experience to be collected by the agent. In practical settings, such as robotics, this involves repeatedly attempting a task, resetting the environment between each attempt. However, not all tasks are easily or automatically reversible. In practice, this learning process requires extensive human intervention. In this work, we propose an autonomous method for safe and efficient reinforcement learning that simultaneously learns a forward and reset policy, with the reset policy resetting the environment for a subsequent attempt. By learning a value function for the reset policy, we can automatically determine when the forward policy is about to enter a non-reversible state, providing for uncertainty-aware safety aborts. Our experiments illustrate that proper use of the reset policy can greatly reduce the number of manual resets required to learn a task, can reduce the number of unsafe actions that lead to non-reversible states, and can automatically induce a curriculum.[1]

## 1 INTRODUCTION

Deep reinforcement learning (RL) algorithms have the potential to automate acquisition of complex behaviors in a variety of real-world settings. Recent results have shown success on games (Mnih et al. (2013)), locomotion (Schulman et al. (2015)), and a variety of robotic manipulation skills (Pinto & Gupta (2017); Schulman et al. (2016); Gu et al. (2017)). However, the complexity of tasks achieved with deep RL in simulation still exceeds the complexity of the tasks learned in the real world. Why have real-world results lagged behind the simulated accomplishments of deep RL algorithms?

One challenge with real-world application of deep RL is the scaffolding required for learning: a bad policy can easily put the system into an unrecoverable state from which no further learning is possible. For example, an autonomous car might collide at high speed, and a robot learning to clean glasses might break them. Even in cases where failures are not catastrophic, some degree of human intervention is often required to reset the environment between attempts (e.g., Chebotar et al. (2017)).

Most RL algorithms require sampling from the initial state distribution at the start of each episode. On real-world tasks, this operation often corresponds to a manual reset of the environment after every episode, an expensive solution for complex environments. Even when tasks are designed so that these resets are easy (e.g., Levine et al. (2016) and Gu et al. (2017)), manual resets are necessary when the robot or environment breaks (e.g., Gandhi et al. (2017)). The bottleneck for learning many real-world tasks is not that the agent collects data too slowly, but rather that data collection stops entirely when the agent is waiting for a manual reset. To avoid manual resets caused by the environment breaking, task designers often add negative rewards to dangerous states and intervene to prevent agents from taking dangerous actions. While this works well for simple tasks, scaling to more complex environments requires writing large numbers of rules for types of actions the robot should avoid. For example, a robot should avoid hitting itself, except when clapping. One interpretation of

---

[∗]Work done as a member of the Google AI Residency Program (g.co/brainresidency)

[1]Videos of our experiments: https://sites.google.com/site/mlleavenotrace/

our method is as automatically learning these safety rules. Decreasing the number of manual resets required to learn to a task is important for scaling up RL experiments outside simulation, allowing researchers to run longer experiments on more agents for more hours.

We propose to address these challenges by forcing our agent to "leave no trace." The goal is to learn not only how to do the task at hand, but also how to undo it. The intuition is that the *sequences of actions that are reversible are safe*; it is always possible to undo them to get back to the original state. This property is also desirable for continual learning of agents, as it removes the requirements for manual resets. In this work, we learn two policies that alternate between attempting the task and resetting the environment. By learning how to reset the environment at the end of each episode, the agent we learn requires significantly fewer manual resets. Critically, our value-based reset policy restricts the agent to only visit states from which it can return, intervening to prevent the forward policy from taking potentially irreversible actions. Using the reset policy to regularize the forward policy encodes the assumption that whether our learned reset policy can reset is a good proxy for whether any reset policy can reset. The algorithm we propose can be applied to both deterministic and stochastic MDPs. For stochastic MDPs we say that an action is reversible if the probability that an oracle reset policy can successfully reset from the next state is greater than some safety threshold. The set of states from which the agent knows how to return grows over time, allowing the agent to explore more parts of the environment as soon as it is safe to do so.

The main contribution of our work is a framework for continually and jointly learning a reset policy in concert with a forward task policy. We show that this reset policy not only automates resetting the environment between episodes, but also helps ensure safety by reducing how frequently the forward policy enters unrecoverable states. Incorporating uncertainty into the value functions of both the forward and reset policy further allows us to make this process risk-aware, balancing exploration against safety. Our experiments illustrate that this approach reduces the number of "hard" manual resets required during learning of a variety of simulated robotic skills.

## 2 RELATED WORK

Our method builds off previous work in areas of safe exploration, multiple policies, and automatic curriculum generation. Previous work has examined safe exploration in small MDPs. Moldovan & Abbeel (2012a) examine risk-sensitive objectives for MDPs, and propose a new objective of which minmax and expectation optimization are both special cases. Moldovan & Abbeel (2012b) consider safety using ergodicity, where an action is safe if it is still possible to reach every other state after having taken that action. These methods are limited to small, discrete MDPs where exact planning is straightforward. Our work includes a similar notion of safety, but can be applied to solve complex, high-dimensional tasks. Thomas et al. (2015a;b) prove high confidence bounds for off policy evaluation and policy improvement. While these works look at safety as guaranteeing some reward, our work defines safety as guaranteeing that an agent can reset.

Previous work has also used multiple policies for safety and for learning complex tasks. Han et al. (2015) learn a sequence of forward and reset policies to complete a complex manipulation task. Similar to Han et al. (2015), our work learns a reset policy to undo the actions of the forward policy. While Han et al. (2015) engage the reset policy when the forward policy fails, we preemptively predict whether the forward policy will fail, and engage the reset policy before allowing the forward policy to fail. Similar to our approach, Richter & Roy (2017) also propose to use a safety policy that can trigger an "abort" to prevent a dangerous situation. However, in contrast to our approach, Richter & Roy (2017) use a heuristic, hand-engineered reset policy, while our reset policy is learned simultaneously with the forward policy. Kahn et al. (2017) uses uncertainty estimation via bootstrap to provide for safety. Our approach also uses bootstrap for uncertainty estimation, but unlike our method, Kahn et al. (2017) does not learn a reset or safety policy.

Learning a reset policy is related to curriculum generation: the reset controller is engaged in increasingly distant states, naturally providing a curriculum for the reset policy. Prior methods have studied curriculum generation by maintaining a separate goal setting policy or network (Sukhbaatar et al., 2017; Matiisen et al., 2017; Held et al., 2017). In contrast to these methods, we do not set explicit goals, but only allow the reset policy to abort an episode. When learning the forward and reset policies jointly, the training dynamics of our reset policy resemble those of reverse curriculum generation (Florensa et al., 2017), but in reverse. In particular, reverse curriculum learning can be

viewed as a special case of our method: our reset policy is analogous to the learner in the reverse curriculum, while the forward policy plays a role similar to the initial state selector. However, reverse curriculum generation requires that the agent can be reset to any state (e.g., in a simulator), while our method is specifically aimed at streamlining real-world learning, through the use of uncertainty estimation and early aborts.

## 3  PRELIMINARIES

In this section, we discuss the episodic RL problem setup, which motivates our proposed joint learning of forward and reset policies. RL considers decision-making problems that consist of a state space $\mathcal{S}$, action space $\mathcal{A}$, transition dynamics $P(s' \mid s, a)$, an initial state distribution $p_0(s)$, and a scalar reward function $r(s, a)$. In episodic, finite horizon tasks, the objective is to find the optimal policy $\pi^*(a \mid s)$ that maximizes the expected sum of $\gamma$-discounted returns, $\mathbb{E}_\pi \left[ \sum_{t=0}^{T} \gamma^t r(s_t, a_t) \right]$, where $s_0 \sim p_0$, $a_t \sim \pi(a_t \mid s_t)$, and $s_{t+1} \sim P(s_{t+1} \mid s_t, a_t)$.

Typical RL training routines involve iteratively sampling new episodes; at the end of each episode, a new starting state $s_0$ is sampled from a given initial state distribution $p_0$. In practical applications, such as robotics, this procedure involves a hard-coded reset policy or a human intervention to manually reset the agent. Our work is aimed at avoiding these manual resets by learning an additional reset policy that satisfies the following property: when the reset policy is executed from any state reached by the forward policy, the distribution over final states is close to the initial state distribution $p_0$. If we learn such a reset policy, then the agent never requires querying the black-box distribution $p_0$ and can continually learn on its own.

## 4  CONTINUAL LEARNING WITH JOINT FORWARD-RESET POLICIES

Our method for continual learning relies on jointly learning a *forward policy* and *reset policy*, using *early aborts* to avoid manual resets. The forward policy aims to maximize the task reward, while the reset policy takes actions to reset the environment. Both have the same state and action spaces, but are given different reward objectives. The forward policy reward $r_f(s, a)$ is the usual task reward given by the environment. The reset policy reward $r_r(s)$ is designed to approximate the initial state distribution. In practice, we found that a very simple design worked well for our experiments. We used the negative distance to some start state, plus any reward shaping included in the forward reward.

To make this set-up applicable for solving the task, we make two assumptions on the task environment. First, we make the weak assumption that there exists a policy that can reset from at least one of the reachable states with maximum reward in the environment. This assumption ensures that it is possible to solve the task without any manual resets. Many manipulation and locomotion tasks in robotics satisfy this assumption. As a counterexample, the Atari game Ms. Pacman violates this assumption because transitioning from one level to the next level is not reversible; the agent cannot transition from level 3 back to level 1. Second, we assume that the initial state distribution is unimodal and has narrow support. This assumption ensures that the distribution over the reset policy's final state is close to the initial state distribution $p_0$. If the initial state distribution were multi-modal, the reset policy might only learn to return to one of these modes. Detecting whether an environment violates this second assumption is straightforward. A mismatch between $p_0$ and the reset policy's final state distribution will cause the forward policy to earn a small reward when the initial state is sampled from $p_0$ and a larger reward when the initial state is the final state of the reset policy.

We choose off-policy actor-critic as the base RL algorithm (Silver et al., 2014; Lillicrap et al., 2015), since its off-policy learning allows sharing of the experience between the forward and reset policies. Additionally, the Q-functions can be used to signal early aborts. Our method can also be used directly with any other Q-learning method (Watkins & Dayan, 1992; Mnih et al., 2013; Gu et al., 2017; Amos et al., 2016; Metz et al., 2017).

### 4.1  EARLY ABORTS

The reset policy learns how to transition from the forward policy's final state back to an initial state. In challenging domains where the reset policy is unable to reset from some states or would

take prohibitively long to reset, a costly manual reset is required. The reset policy offers a natural mechanism for reducing these manual resets. We observe that, for states from which we cannot quickly reset, the value function of the reset policy will be low. We can therefore use this value function (or, specifically, its Q-function) as a metric to determine when to terminate the forward policy, performing an *early abort*.

Before an action proposed by the forward policy is executed in the environment, it must be "approved" by the reset policy. In particular, if the reset policy's Q-value for the proposed action is too small, then an early abort is performed: the proposed action is not taken and the reset policy takes control. Formally, early aborts restrict exploration to a 'safe' subspace of the MDP. Let $\mathcal{E} \subseteq \mathcal{S} \times \mathcal{A}$ be the set of (possibly stochastic) transitions, and let $Q_{reset}(s, a)$ be the Q-value of our reset policy at state $s$ taking action $a$. The subset of transitions $\mathcal{E}^* \in \mathcal{E}$ allowed by our algorithm is

$$\mathcal{E}^* \triangleq \{(s, a) \in \mathcal{E} \mid Q_{reset}(s, a) > Q_{min}\} \tag{1}$$

Noting that $V(s) \triangleq \max_{a \in \mathcal{A}} Q(s, a)$, we see that given access to the true Q-values, Leave No Trace only visits safe states:

$$\mathcal{S}^* \triangleq \{s \mid (s, a) \in \mathcal{E}^* \text{ for at least one } a \in \mathcal{A}\} \tag{2}$$
$$= \{s \mid V_{reset}(s) > Q_{min}\} \tag{3}$$

In Appendix A, we prove that if we learn the true Q-values for the reset policy, then early aborts restrict the forward policy to visiting states that are safe in expectation at convergence.

Early aborts can be interpreted as a learned, dynamic, safety constraint, and a viable alternative for the manual constraints that are typically used for real-world RL experiments. Early aborts promote safety by preventing the agent from taking actions from which it cannot recover. These aborts are dynamic because the states at which they occur change throughout training as more states are considered safe. Early aborts can make learning the forward policy easier by preventing the agent from entering unsafe states. We experimentally analyze early aborts in Section 6.3, and discuss how our approach handles over/under-estimates of Q-values in Appendix B.

### 4.2 HARD RESETS

A *hard reset* is an action that resamples that state from the initial state distribution. Hard resets are available to an external agent (e.g., a human) but not the learned agent. Early aborts decrease the requirement for "hard" resets, but do not eliminate them, since an imperfect reset policy might still miss a dangerous state early in the training process.

It is challenging to identify whether *any* policy can reset from the current state. Formally, we define a set of states $\mathcal{S}_{reset}$ that give a reward greater than $r_{min}$ to the reset policy:

$$\mathcal{S}_{reset} \triangleq \{s \mid r_r(s) > r_{min}\} \tag{4}$$

We say that we are in an irreversible state if we have not visited a state in $\mathcal{S}_{reset}$ within the past $N$ episodes, where $N$ is a hyperparameter. This is a necessary but not sufficient condition, as the reset policy may have not yet learned to reset from a safe state. Increasing $N$ decreases the number of hard resets. However, when we are in an irreversible state, increasing $N$ means that we remain in that state (learning nothing) for more episodes. Section 6.4 empirically examines this trade-off. In practice, the setting of this parameter should depend on the cost of hard resets.

### 4.3 ALGORITHM SUMMARY

Our full algorithm (Algorithm 1) consists of alternately running a forward policy and reset policy. When running the forward policy, we perform an early abort if the Q-value for the reset policy is less than $Q_{min}$. Only if the reset policy fails to reset after $N$ episodes do we do a manual reset.

### 4.4 VALUE FUNCTION ENSEMBLES

The accuracy of the Q-value estimates directly affects task reward and indirectly affects safety (through early aborts). Our Q-values may not be good estimates of the true value function for

---

**Algorithm 1** Joint Training

---
1: **repeat**
2:     **for** `max_steps_per_episode` **do**
3:         $a \leftarrow$ FORWARD_AGENT.CHOOSE_ACTION($s$)
4:         **if** RESET_AGENT.Q($s, a$) $< Q_{min}$ **then**
5:             Switch to reset policy.                                 ▷ Early Abort
6:         $(s, r) \leftarrow$ ENVIRONMENT.STEP($a$)
7:         Update the forward policy.
8:     **for** `max_steps_per_episode` **do**
9:         $a \leftarrow$ RESET_AGENT.CHOOSE_ACTION($s$)
10:         $(s, r) \leftarrow$ ENVIRONMENT.STEP($a$)
11:         Update the reset policy.
12:     Let $\mathcal{S}_{reset}^{N}$ be the final states from the last $N$ reset episodes.
13:     **if** $\mathcal{S}_{reset}^{N} \cap \mathcal{S}_{reset} = \emptyset$ **then**                     ▷ Detect Failed Reset (Eq. 4)
14:         $s \leftarrow$ ENVIRONMENT.RESET()                       ▷ Hard Reset

---

previously-unseen states. To address this, we train Q-functions for both the forward and reset policies that provide uncertainty estimates. Several prior works have explored how uncertainty estimates can be obtained in such settings (Gal & Ghahramani, 2016; Osband et al., 2016). In our method, we train an ensemble of Q-functions, each with a different random initialization. This technique has been established in the literature as a principled way to provides a distribution over Q-values at each state given the observed data Osband et al. (2016); Chen et al. (2017).

Given this distribution over Q-values, we can propose three strategies for early aborts:

> *Optimistic Aborts:* Perform an early abort only if all the Q-values are less than $Q_{min}$. Equivalently, do an early abort if $\max_\theta Q_{reset}^\theta(s, a) < Q_{min}$.
>
> *Realist Aborts:* Perform an early abort if the mean Q-value is less than $Q_{min}$.
>
> *Pessimistic Aborts:* Perform an early abort if any of the Q-values are less than $Q_{min}$. Equivalently, do an early abort if $\min_\theta Q_{reset}^\theta(s, a) < Q_{min}$.

We expect that optimistic aborts will provide better exploration at the cost of more hard resets, while pessimistic aborts should decrease hard resets, but may be unable to effectively explore. We empirically test this hypothesis in Appendix C.

## 5 SMALL-SCALE DIDACTIC EXAMPLE

We first present a small didactic example to illustrate how our forward and reset policies interact and how cautious exploration reduces the number of hard resets. We first discuss the grid-world in Figure 1. The states with red borders are absorbing, meaning that the agent cannot leave them and must use a hard reset. The agent receives a reward of 1 for reaching the goal state, and 0 otherwise. States are colored based on the number of early aborts triggered in each state. Note that most aborts occur next to the initial state, when the forward policy attempts to enter the absorbing state South-East of the start state, but is blocked by the reset policy. In Figure 2, we present a harder environment, where the task can be successfully completed by reaching one of the two goals, exactly one of which is reversible. The forward policy has no preference for which goal is better, but the reset policy successfully prevents the forward policy from entering the absorbing goal state, as indicated by the much larger early abort count in the blue-colored state next to the absorbing goal.

Figure 3 shows how changing the early abort threshold to explore more cautiously reduces the number of failures. Increasing $Q_{min}$ from 0 to 0.4 reduced the number of hard resets by

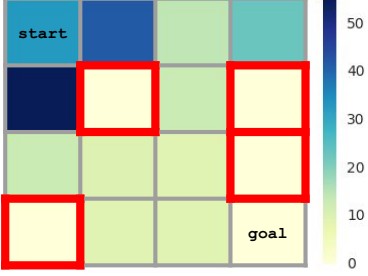

Figure 1: Early aborts in gridworld.

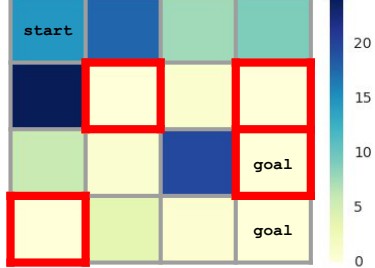

Figure 2: Early aborts with an absorbing goal.

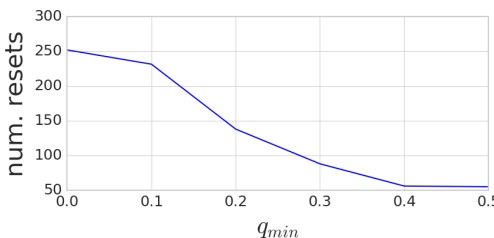

Figure 3: Early abort threshold: In our didactic example, increasing the early abort threshold causes more cautious exploration (left) without severely increasing the number of steps to solve (right).

78% without increasing the number of steps to solve the task. In a real-world setting, this might produce a substantial gain in efficiency, as time spend waiting for a hard reset could be better spent collecting more experience. Thus, for some real-world experiments, increasing $Q_{min}$ can decrease training time even if it requires more steps to learn.

# 6  CONTINUOUS ENVIRONMENT EXPERIMENTS

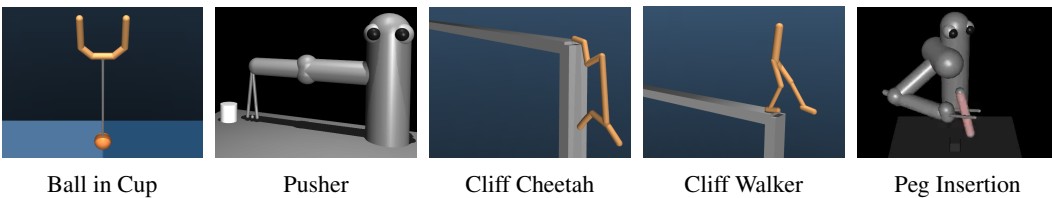

In this section, we use the five complex, continuous control environments shown above to answer questions about our approach. While ball in cup and peg insertion are completely reversible, the other environments are not: the pusher can knock the puck outside its workspace and the cheetah and walker can jump off a cliff. Crucially, reaching the goal states or these irreversible states does not terminate the episode, so the agent remains in the irreversible state until it calls for a hard reset. To ensure fair evaluation of all approaches, we use a different procedure for evaluation than for training. We evaluate the performance of a policy by creating a copy of the policy in a separate thread, running the *forward* policy for a fixed number of steps, and computing the average per-step reward. All approaches observe the same amount of data during training. We visualize the training dynamics and provide additional plots and experimental details are in the Appendix.

## 6.1  WHY LEARN A RESET CONTROLLER?

One proposal for learning without resets is to run the forward policy until the task is learned. This "forward-only" approach corresponds to the standard, fully online, non-episodic lifelong RL setting, commonly studied in the context of temporal difference learning (Sutton & Barto (1998)). We show that this approach fails, even on reversible environments where safety is not a concern. We benchmarked the forward-only approach and our method on ball in cup, using no hard resets for either. Figure 5 shows that our approach solves the task while the "forward-only" approach fails to learn how to catch the ball when initialized below the cup. Note that the x axis includes steps taken by the reset policy. Once the forward-only approach catches the ball, it gets maximum reward by keeping the ball in the cup. In contrast, our method learns to solve this task by automatically resetting the environment after each attempt, so the forward policy can practice catching the ball without hard resets. As an upper bound,

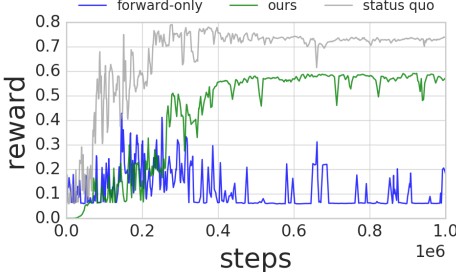

Figure 5: We compare our method to a non-episodic ("forward-only") approach on ball in cup. Although neither uses hard resets, only our method learns to catch the ball. As an upper bound, we also show the "status quo" approach that performs a hard reset after episode, which is often impractical outside simulation.

we show policy reward for the "status quo" approach, which performs a hard reset after every attempt. Note that the dependence on hard resets makes this third method impractical outside simulation.

## 6.2 DOES OUR METHOD REDUCE MANUAL RESETS?

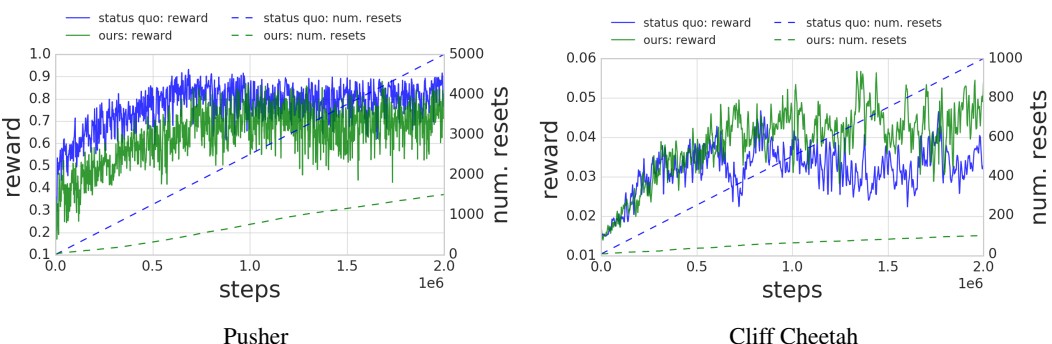

Figure 6: Our method achieves equal or better rewards than the status quo with fewer manual resets.

Our first goal is to reduce the number of hard resets during learning. In this section, we compare our algorithm to the standard, episodic learning setup ("status quo"), which only learns a forward policy. As shown in Figure 6 (left), the conventional approach learns the pusher task somewhat faster than ours, but our approach eventually achieves the same reward with half the number of hard resets. In the cliff cheetah task (Figure 6 (right)), not only does our approach use an order of magnitude fewer hard resets, but the final reward of our method is substantially higher. This suggests that, besides reducing the number of resets, the early aborts can actually aid learning by preventing the forward policy from wasting exploration time waiting for resets in irreversible states.

## 6.3 DO EARLY ABORTS AVOID HARD RESETS?

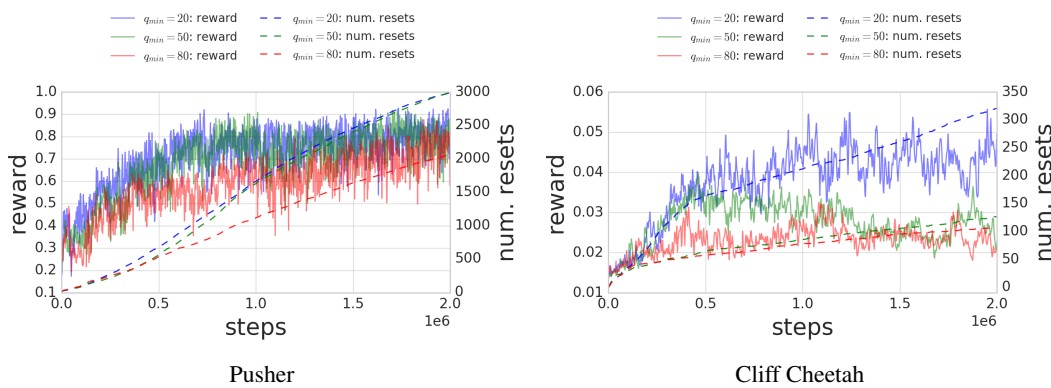

Figure 7: Early abort threshold: Increasing the early abort threshold to act more cautiously avoids many hard resets, indicating that early aborts help avoid irreversible states.

To test whether early aborts prevent hard resets, we can see if the number of hard resets increases when we lower the early abort threshold. Figure 7 shows the effect of three values for $Q_{min}$ while learning the pusher and cliff cheetah. In both environments, decreasing the early abort threshold increased the number of hard resets, supporting our hypothesis that early aborts prevent hard resets. On pusher, increasing $Q_{min}$ to 80 allowed the agent to learn a policy that achieved nearly the same reward using 33% fewer hard resets. The cliff cheetah task has lower rewards than pusher, even an early abort threshold of 10 is enough to prevent 69% of the total early aborts that the status quo would have performed.

## 6.4 MULTIPLE RESET ATTEMPTS

While early aborts help avoid hard resets, our algorithm includes a mechanism for requesting a manual reset if the agent reaches an unresettable state. As described in Section 4.2, we only perform a hard reset if the reset agent fails to reset in $N$ consecutive episodes. Figure 8 shows how the number of reset attempts, $N$, affects hard resets and reward. On the pusher task, when our algorithm was given a single reset attempt, it used 64% fewer hard resets than the status quo approach would have. Increasing the number of reset attempts to 4 resulted in another 2.5x reduction in hard resets, while decreasing the reward by less than 25%. On the cliff cheetah task, increasing the number of reset attempts brought the number of resets down to nearly zero, without changing the reward. Surprisingly, these results indicate that for some tasks, it is possible to learn an equally good policy with significantly fewer hard resets.

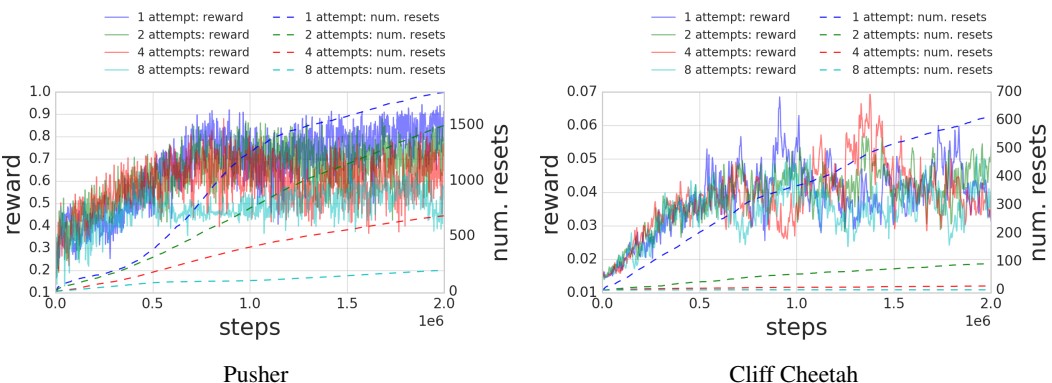

Figure 8: Reset attempts: Increasing the number of reset attempts reduces hard resets. Allowing too many reset attempts reduces reward for the pusher environment.

## 6.5 ENSEMBLES ARE SAFER

Our approach uses an ensemble of value functions to trigger early aborts. Our hypothesis was that our algorithm would be sensitive to bias in the value function if we used a single Q network. To test this hypothesis, we varied the ensemble size from 1 to 50. Figure 9 shows the effect on learning the pushing task. An ensemble with one network failed to learn, but still required many hard resets. Increasing the ensemble size slightly decreased the number of hard resets without affecting the reward.

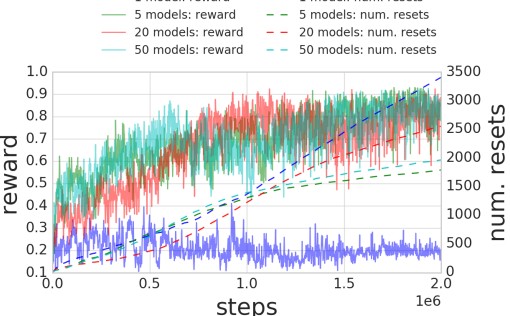

Figure 9: Increasing ensemble size boosts policy reward while decreasing rate of hard resets.

## 6.6 AUTOMATIC CURRICULUM LEARNING

Our method can automatically produce a curriculum in settings where the desired skill is performed by the reset policy, rather than the forward policy. As an example, we evaluate our method on a peg insertion task, where the reset policy inserts the peg and the forward policy removes it. The reward for a successful peg insertion is provided only when the peg is in the hole, making this task challenging to learn with random exploration. Hard resets provide illustrations of what a successful outcome looks like, but do not show how to achieve it. Our algorithm starts with the peg in the hole and runs the forward (peg removal) policy until an early abort occurs. As the reset (peg

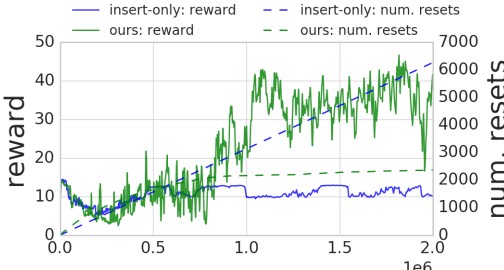

Figure 10: Our method automatically induces a curriculum, allowing the agent to solve peg insertion with sparse rewards.

insertion) policy improves, early aborts occur further and further from the hole. Thus, the initial state distribution for the reset (peg insertion) policy moves further and further from the hole, increasing the difficulty of the task as the policy improves. We compare our approach to an "insert-only" baseline that only learns the peg insertion policy – we manually remove the peg from the hole after every episode. For evaluation, both approaches start outside the hole. Figure 10 shows that only our method solves the task. The number of resets required by our method plateaus after one million steps, indicating that it has solved the task and no longer requires hard resets at the end of the episode. In contrast, the "insert-only" baseline fails to solve the task, never improving its reward. Thus, even if reducing manual resets is not important, the curriculum automatically created by Leave No Trace can enable agents to learn policies they otherwise would be unable to solve.

## 7 CONCLUSION

In this paper, we presented a framework for automating reinforcement learning based on two principles: automated resets between trials, and early aborts to avoid unrecoverable states. Our method simultaneously learns a forward and reset policy, with the value functions of the two policies used to balance exploration against recoverability. Experiments in this paper demonstrate that our algorithm not only reduces the number of manual resets required to learn a task, but also learns to avoid unsafe states and automatically induces a curriculum.

Our algorithm can be applied to a wide range of tasks, only requiring a few manual resets to learn some tasks. During the early stages of learning we cannot accurately predict the consequences of our actions. We cannot learn to avoid a dangerous state until we have visited that state (or a similar state) and experienced a manual reset. Nonetheless, reducing the number of manual resets during learning will enable researchers to run experiments for longer on more agents. A second limitation of our work is that we treat all manual resets as equally bad. In practice, some manual resets are more costly than others. For example, it is more costly for a grasping robot to break a wine glass than to push a block out of its workspace. An approach not studied in this paper for handling these cases would be to specify costs associated with each type of manual reset, and incorporate these reset costs into the learning algorithm.

While the experiments for this paper were done in simulation, where manual resets are inexpensive, the next step is to apply our algorithm to real robots, where manual resets are costly. A challenge introduced when switching to the real world is automatically identifying when the agent has reset. In simulation we can access the state of the environment directly to compute the distance between the current state and initial state. In the real world, we must infer states from noisy sensor observations to deduce if they are the same. If we cannot distinguish between the state where the forward policy started and the state where the reset policy ended, then we have succeeded in Leaving No Trace!

**Acknowledgements:** We thank Sergio Guadarrama, Oscar Ramirez, and Anoop Korattikara for implementing DDPG and thank Peter Pastor for insightful discussions.

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

## A  SAFETY INVARIANT PROOF

In this section, we prove that if we indeed learn the true Q-values for the reset policy, then the abort condition stipulated by our method will keep the forward policy safe (able to reset) for deterministic infinite-horizon discounted reward MDPs. For stochastic MDPs, the abort condition will keep the forward policy safe in expectation. Thus, the abort condition is effective at convergence. Before convergence, the reliability of this abort condition depends on the accuracy of the learned Q-values. In practice, we partially mitigate issues with imperfect Q functions by means of the Q-function ensemble (Section 4.4).

### A.1  ASSUMPTIONS

In the proofs that follow, we make the following assumptions:

1. *The reward function for the reset policy depends only on the current state.* We use $r_r(s)$ as the reset reward received for arriving at state $s$ throughout this proof.

2. *From every state $s_t$, there exists an action $a_t$ such that the expected reset reward at the next state is at least as large as the reset reward at the current state:*

$$\mathbb{E}_{s_{t+1} \sim p(s_{t+1}|s_t, a_t)}[r_r(s_{t+1})] \geq r_r(s_t)$$

For example, if the reset reward is uniform over $\mathcal{S}_{reset}$ and zero everywhere else, then this assumption requires that for every state $\mathcal{S}_{reset}$, there exists an action that deterministically transitions to another state in $\mathcal{S}_{reset}$. As a counterexample, a modified cliff cheetah environment where the cheetah is initialized a meter above the ground does not satisfy this assumption. If the cheetah in $\mathcal{S}_{reset}$ (above the ground), there are no actions it can take to stop itself from falling to the ground and leaving $\mathcal{S}_{reset}$.

### A.2  PROOFS

In the proofs below, we assume a stochastic MDP. For a deterministic MDP, we can remove the expectations over $s_{t+1}$ (Lemma 4). To begin, we use the two assumptions to establish a lower bound for the value of any state for the reset policy.

**Lemma 1.** *For every state $s \in \mathcal{S}$, the expected cumulative discounted reward for the reset agent is greater than a term that depends on the discount $\gamma$ and the reward of the current state $r_r(s)$:*

$$V_{reset}(s) \geq \frac{1}{1-\gamma} r_r(s) \qquad \forall\, s \in \mathcal{S} \tag{5}$$

*Proof.* As a consequence of the assumptions, a reset policy at state $s$ that acts optimally is guaranteed to receive an expected reset reward of at least $r_r(s)$ in every future time step. Thus, its expected cumulative discounted reward is at least $\frac{1}{1-\gamma} r_r(s)$. $\qquad\square$

Next, we show that the reset policy can choose actions so that the Q-values do not decrease in expectation.

**Theorem 1.** *For any state $s_t \in \mathcal{S}$ and action $a_t \in \mathcal{A}$, there exists another action $a_{t+1}^* \in \mathcal{A}$ such that*

$$\mathbb{E}_{s_{t+1} \sim p(s_{t+1}|s_t, a_t)} \left[ Q_{reset}(s_{t+1}, a_{t+1}^*) \right] \geq Q_{reset}(s_t, a_t)$$

In the proof that follows, note that the next state $s_{t+1}$ is an unknown random variable. Functions of $s_{t+1}$ are also random variables.

*Proof.* Let state $s_t$ and action $a_t$ be given and let $s_{t+1}$ be a random variable indicating the next state following a possibly stochastic transition. Let $a_{t+1}^*$ to be the action with largest Q-value at the next state:

$$a_{t+1}^* = \arg\max_{a_{t+1}} Q(s_{t+1}, a_{t+1})$$

We want to bound the expected difference between $Q_{reset}(s_t, a_t)$ and $Q_{reset}(s_{t+1}, a_{t+1}^*)$, where the expectation is with respect to the unknown next state $s_{t+1}$. We begin by unrolling the first term of the Q-value. Because $a_{t+1}^*$ is defined to be the action with largest Q-value, we can replace the first Q-value expression with the value function. We then apply the bound from Lemma 1. For brevity, we omit the subscript $reset$ and omit that the expectation is over $s_{t+1}$.

$$
\begin{aligned}
\mathbb{E}[Q(s_{t+1}, a_{t+1}^*) - Q(s_t, a_t)] &= \mathbb{E}\left[Q(s_{t+1}, a_{t+1}^*) - (r_r(s_{t+1}) + \gamma V(s_{t+1}))\right] \\
&= \mathbb{E}\left[V(s_{t+1}) - r_r(s_{t+1}) - \gamma V(s_{t+1}))\right] \\
&= \mathbb{E}\left[(1 - \gamma)V(s_{t+1}) - r_r(s_{t+1}))\right] \\
&\geq \mathbb{E}\left[(1 - \gamma)\frac{1}{1 - \gamma}r_r(s_{t+1}) - r_r(s_{t+1})\right] \\
&= \mathbb{E}\left[r_r(s_{t+1}) - r_r(s_{t+1})\right] \\
&= 0
\end{aligned}
$$

$\square$

Next, we want to show that if one state is safe, the next state will also be safe in expectation. As a reminder, we say transitions $(s, a)$ in $\mathcal{E}^*$ are safe and states $s$ in $\mathcal{S}^*$ are safe (Eq. 1 and 3):

$$
\mathcal{E}^* \triangleq \{(s, a) \in \mathcal{E} \mid Q_{reset}(s, a) > Q_{min}\}
$$
$$
\mathcal{S}^* \triangleq \{s \mid (s, a) \in \mathcal{E}^* \text{ for at least one } a \in \mathcal{A}\}
$$

**Lemma 2.** *Let safe state $s_t \in \mathcal{S}^*$ be given and choose an action $a_t$ such that $(s_t, a_t) \in \mathcal{E}^*$. Then the following state $s_{t+1}$ is also safe in expectation:*

$$
\mathbb{E}_{s_{t+1} \sim p(s_{t+1}|s_t, a_t)}\left[Q_{reset}(s_{t+1}, a_{t+1}^*)\right] \geq Q_{min}
$$

*Proof.* By our assumption that $(s_t, a_t) \in \mathcal{E}^*$, we know $Q_{reset}(s_t, a_t) > Q_{min}$. Combining with Theorem 1, we get

$$
\mathbb{E}_{s_{t+1} \sim p(s_{t+1}|s_t, a_t)}\left[Q_{reset}(s_{t+1}, a_{t+1}^*)\right] \geq Q_{min}
$$

Thus, state $s_{t+1}$ is safe in expectation. $\square$

Finally, we want to show that Leave No Trace only visits safe states in expectation.

**Lemma 3.** *If the initial state $s_0$ is safe, then Leave No Trace only visits states that are also safe in expectation.*

*Proof.* Proof by induction. We assumed that the initial state $s_0$ is safe. Lemma 2 shows that safety is a preserved invariant. Thus, each future state $s_t$ is also safe in expectation. $\square$

Leave No Trace being safe in expectation means that if we look an arbitrary number of steps into the future, the expected Q-value for that state is at least $Q_{min}$. Equivalently, the probability that the state we arrive at is safe is greater than 50%.

Deterministic MDPs are a special case for which we can prove that Leave No Trace only visits safe states (not in expectation).

**Lemma 4.** *For deterministic MDPs, if the initial state $s_0$ is safe, then Leave No Trace only visits states that are also safe (not in expectation.)*

*Proof.* When the next state $s_{t+1}$ is a deterministic function of the current state $s_t$ and action $a_t$, we can remove the expectation over $s_{t+1}$ from Theorem 1 and Lemma 2. Thus, if the initial state $s_0$ is safe, we are guaranteed that every future state is also safe. $\square$

### A.3 LEAVE NO TRACE IN PRACTICE

In practice, Leave No Trace does visit unsafe states (though significantly less frequently than existing approaches). First, the proofs above only show that each state is safe in expectation. We do not prove that every state is safe with high probability. Second, we do not have access to the true Q-values. Our learned Q-function may overestimate the Q-value of some action, leading us to take an unsafe action. Empirically, we found that using an ensemble of Q-functions helped mitigate this problem, decreasing the number of unsafe actions taken as compared to using a single Q-function (Section 6.5).

## B  Q-VALUE ESTIMATION ERRORS

We introduced early aborts in Section 4.1 and analyzed them in Appendix A under the assumption that we had access to the true Q-values. In practice, our learned Q-values may over/under-estimate the Q-value for the reset policy. First, consider the case that the Q-function overestimates the reset Q-value for state $s_u$, so the agent mistakenly thinks that unsafe state $s_u$ is safe. The agent will visit state $s_u$ and discover that it cannot reset. When the reset Q-function is updated with this experience, it will decrease its predicted reset Q-value for state $s_u$. Second, consider the case that the Q-function underestimates the reset Q-value for state $s_s$, so the agent mistakenly thinks that safe state $s_s$ is unsafe. For continuous tasks, once the agent learns to reset from a nearby safe state, generalization of the Q-function across states will lead the reset policy to assume that it can also reset from the state $s_s$. For discrete state tasks where the Q-function does not generalize across states, we act optimistically in the face of uncertainty by acting based on the largest predicted Q-value from our ensemble, helping to avoid this second case (see Appendix C).

## C  COMBINING AN ENSEMBLE OF VALUE FUNCTIONS

We benchmarked three methods for combining our ensemble of values functions (optimistic, realistic, and pessimistic, as discussed in Section 4.4). Figure 11 compares the three methods on gridworld on the gridworld environment from Section 5. Only the optimistic agent efficiently explored. As expected, the realistic and pessimistic agents, which are more conservative in letting the forward policy continue, fail to explore when $Q_{min}$ is too large.

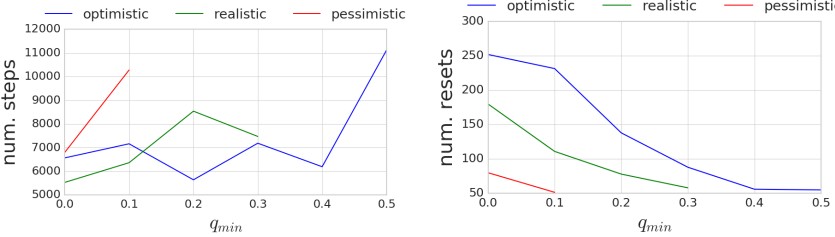

Figure 11: Combining value functions: We compare three methods for ensembling value functions on gridworld. Missing points for the red and green lines indicate that pessimistic and realistic method fail to solve the task for larger values of $Q_{min}$.

Interestingly, for the continuous control environments, the ensembling method makes relatively little difference for the number of resets or final performance, as shown in Figure 12. This suggests that much of the benefit of ensemble comes from its ability to produce less biased abort predictions in novel states, rather than the particular risk-sensitive rule that is used. This result also indicates that no Q-function in the ensemble significantly overestimates or underestimates the value function – such a Q-function would result in bogus Q-value estimates when the ensemble was combined by taking the max or min (respectively).

## D  TRAINING DYNAMICS

In this section, we provide some intuition for the training dynamics of our algorithm. In particular, we visualize the number of steps taken by the forward policy before an early abort occurs. Figure 13

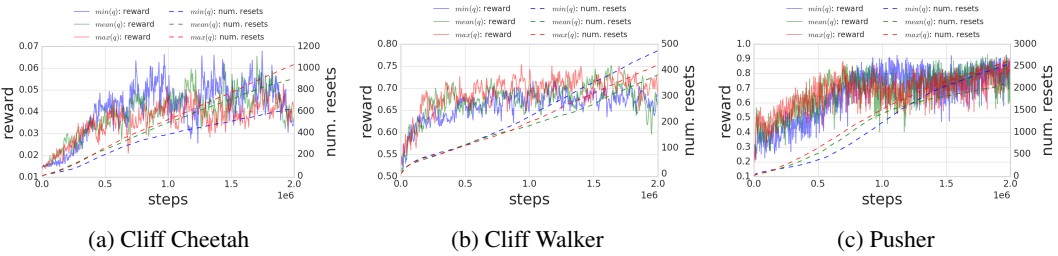

Figure 12: Combining value functions: For continuous environments, the method for combing value functions has little effect.

shows this quantity (the episode length for the forward policy) as a function of training iteration. Note that we stop the forward policy after a fixed number of steps (500 steps for cliff cheetah and cliff walker, 100 steps for pusher) if an early abort has not already occurred. For all tasks, initially the reset policy is unable to reset from any state, so early aborts happen almost immediately. As the reset policy improves, early aborts occur further and further from the initial state distribution, corresponding to longer forward episode lengths. In all tasks, increasing the safety threshold $Q_{min}$ caused early aborts to occur sooner, especially early in training. For the cliff cheetah, another curious pattern emerges when $Q_{min}$ is 10 and 20. After 200 thousand steps, the agent had learned rudimentary policies for running forwards and backwards. As the agent learns to run forwards faster, it reaches the cliff sooner and does an early abort, so the forward episode length actually decreases. For cliff walker, we do not see the same pattern because the forward task is more difficult, so the agent only reaches the cliff near the end of training. Both the cliff walker and pusher environments highlight the sensitivity of our method to $Q_{min}$. If $Q_{min}$ is too small, early aborts will never occur. Automatically tuning the safety threshhold based on the real-world cost of hard resets is an exciting direction for future research.

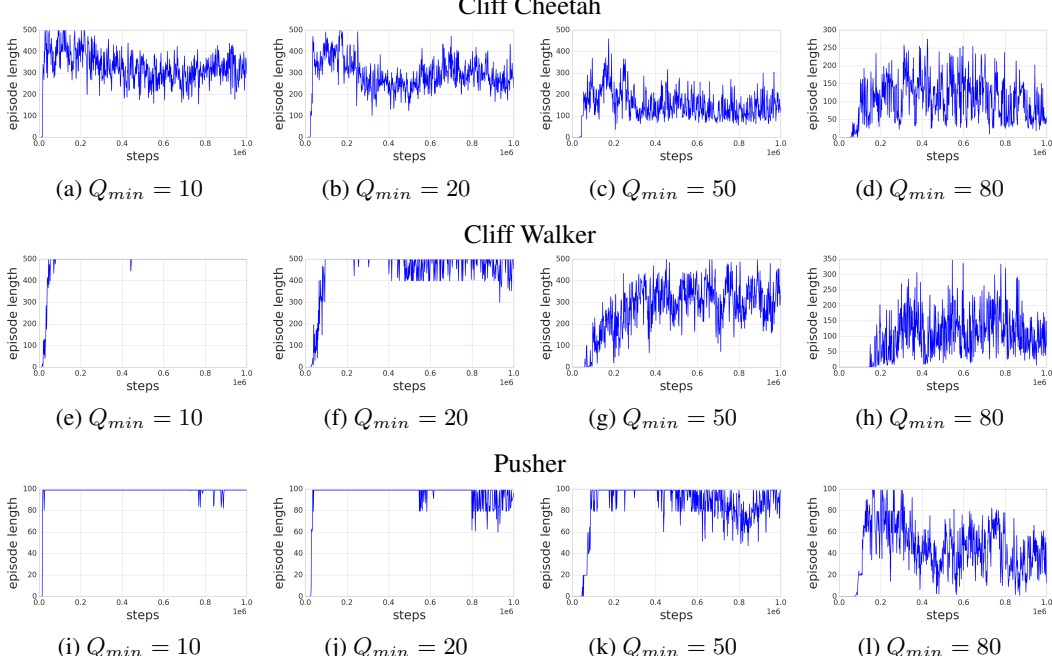

Figure 13: Training dynamics: We show the number of steps taken before an early abort for cliff cheetah (top row), cliff walker (middle row), and pusher (bottom row). Increasing the safety threshold causes early aborts to occur earlier, causing the agent to explore more cautiously. These plots are the average across 5 random seeds.

# E ADDITIONAL FIGURES

For each experiment in the main paper, we chose one or two demonstrative environments. Below, we show all experiments run on cliff cheetah, cliff walker, and pusher.

## E.1 DOES OUR METHOD REDUCE MANUAL RESETS? – MORE PLOTS

This experiment, described in Section 6.2, compared our method to the status quo approach (resetting after every episode). Figure 14 shows plots for all environments.

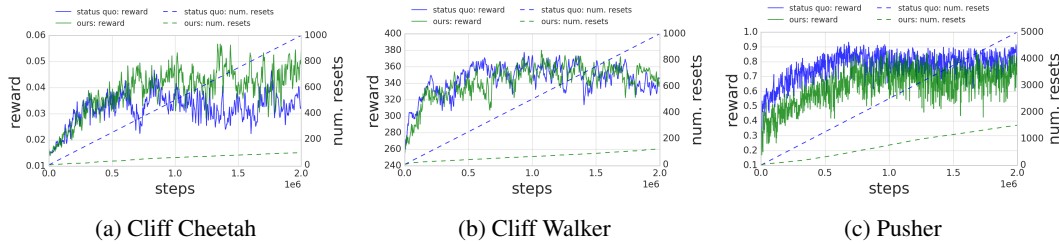

| (a) Cliff Cheetah | (b) Cliff Walker | (c) Pusher |

Figure 14: Experiment from § 6.2

## E.2 DO EARLY ABORTS AVOID HARD RESETS PLOTS? – MORE PLOTS

This experiment, described in Section 6.3, shows the effect of varying the early abort threshold. Figure 15 shows plots for all environments.

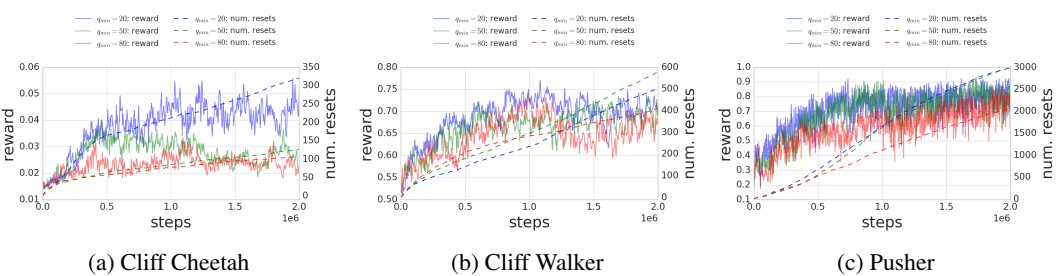

| (a) Cliff Cheetah | (b) Cliff Walker | (c) Pusher |

Figure 15: Experiment from § 6.3

## E.3 MULTIPLE RESET ATTEMPTS – MORE PLOTS

This experiment, described in Section 6.4, shows the effect of increasing the number of reset attempts. Figure 16 shows plots for all environments.

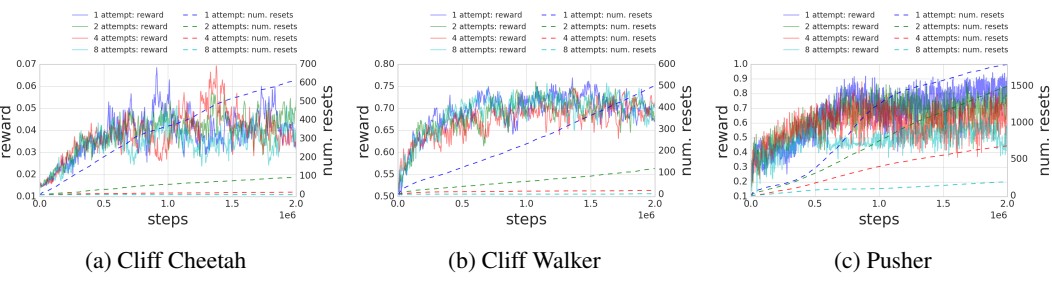

| (a) Cliff Cheetah | (b) Cliff Walker | (c) Pusher |

Figure 16: Experiment from § 6.4

### E.4 Ensembles are Safer – More Plots

This experiment, described in Section 6.5, shows the effect of increasing the number of reset attempts. Figure 17 shows plots for all environments.

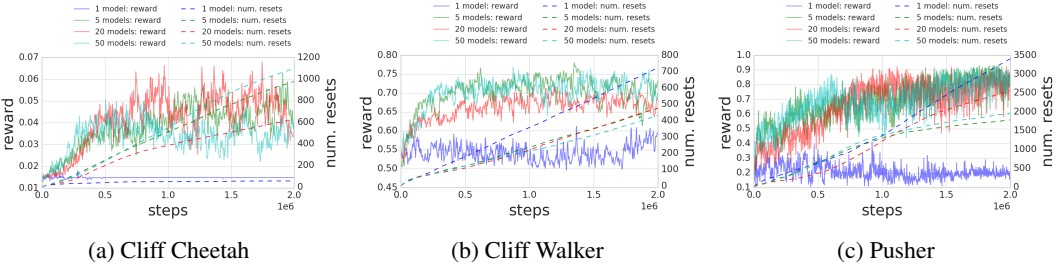

(a) Cliff Cheetah          (b) Cliff Walker          (c) Pusher

Figure 17: Experiment from § 6.5

## F EXPERIMENTAL DETAILS

### F.1 GRIDWORLD EXPERIMENTS

To generate Figures 1 and 2, we averaged early abort counts across 10 random seeds. For Figure 3 we took the median result across 10 random seeds. Both gridworld experiments used 5 models in the ensemble.

### F.2 CONTINUOUS CONTROL ENVIRONMENTS

In this section, we provide additional information on the continuous control environments in our experiments. We use the ball in cup environment implemented in Tassa et al. (2018). The cliff cheetah and cliff walker environments are modified versions of the cheetah and walker environments in Tassa et al. (2018). The pusher environment is a modified version of Pusher-v0 environment in Brockman et al. (2016). Finally, the peg insertion environment is based on Finn et al. (2016).

Below, we provide a high level description of each task, the forward reward, the reset reward, and any reward shaping used. Note that the reset reward is always the Euclidean distance between certain dimensions of the current observation and a reference start observation.

> *Ball in Cup:*
>> *Description:* The agent swings a ball attached by a string up into a cup.
>> *Forward reward:* +1 if the ball is in the cup and 0 otherwise.
>> *Reset reward:* Negative Euclidean distance between current observation and start observation (ball hanging stationary below the cup).
>> *Reward shaping:* Control penalty (negative Euclidean norm of action)
>
> *Cliff Cheetah:*
>> *Description:* The agent learns how to run on a 14m cliff.
>> *Forward reward:* Scaled linear velocity (see Tassa et al. (2018)).
>> *Reset reward:* Negative distance from origin along the X axis (i.e., $-|x|$).
>> *Reward shaping:* Indicator of whether agent is standing and a control penalty (see Tassa et al. (2018)).
>
> *Cliff Walker:*
>> *Description:* The agent learns how to walker on a 6m cliff.
>> *Forward reward:* Scaled linear velocity (see Tassa et al. (2018)).
>> *Reset reward:* Negative distance from origin along the X axis (i.e., $-|x|$).
>> *Reward shaping:* Indicator of whether agent is standing and a control penalty (see Tassa et al. (2018)).

*Pusher:*

> *Description:* The agent pushes a puck to a goal location.
>
> *Forward reward:* Negative Euclidean distance from puck to goal.
>
> *Reset reward:* Negative Euclidean distance from puck to start.
>
> *Reward shaping:* Negative Euclidean distance from end of arm to puck and a control penalty (negative Euclidean norm of action).

*Peg Insertion:*

> *Description:* The agent inserts a peg into a small hole.
>
> *Forward reward:* +1 if the peg is in the hole and 0 otherwise.
>
> *Reset reward:* (Positive) Euclidean distance from peg to hole.
>
> *Reward shaping:* Control penalty (negative Euclidean norm of action).

### F.3 CONTINUOUS CONTROL EXPERIMENTS

We did not do hyperparameter optimization for our experiments, but did run with 5 random seeds. To aggregate results, we took the median number across all random seeds that solved the task. For most experiments, all random seeds solved the task.

For the three continuous control environments, we normalized the rewards to be in $[0, 1]$ so we could use the same hyperparameters for each. The initial state distribution $p_0$ for each task is a uniform distribution centered at some "start pose." Using a discount factor of $\gamma = 0.99$, the cumulative discounted reward was in $[0, 100)$. We defined $\mathcal{S}_{reset}$ as states with reset reward was greater than 0.7.

We used the same DDPG hyperparameters for all continuous control environments:

> *Actor Network:* Two fully connected layers of sizes 400 and 300, with tanh nonlinearities throughout.
>
> *Critic Network:* We apply a 400-dimensional fully connected layer to states, then concatenate the actions and apply another 300-dimensional fully connected layer. Again, we use tanh nonlinearities.

Unless otherwise noted, experiments used an ensemble of size 20, $Q_{min} = 10$, 1 reset attempt, and early aborts using $\min(q)$. The experiments in Section 6.2, our model used 2 reset attempts to better illustrate the potential for our approach to reduce hard resets.

