# OpenReview forum: "Leave no Trace: Learning to Reset for Safe and Autonomous Reinforcement Learning"
_ICLR.cc/2018/Conference — Accept (Poster)_

### Official Review · AnonReviewer1 · 2017-11-28
**A strategy for learning to self-reset in episodic tasks as well as to avoid some types of bad failure**

**Rating:** 6
**Confidence:** 5

**Review:**

If one is committed to doing value-function or policy-based RL for an episodic task on a real physical system, then one has to come up with a way of resetting the domain for new trials.  This paper proposes a good way of doing this:  learn a policy for resetting at the same time as learning a policy for solving the problem.  As a side effect, the Q values associated with the reset policy can be used to predict when the system is about to enter an unrecoverable state and "forbid" the action.

It is, of course, necessary that the domain be, in fact, reversible  (or, at least, that it be possible to reach a starting state from at least one goal state--and it's better if that goal state is not significantly harder to reach than other goal states.

There were a couple of places in the paper that seemed to be to be not strictly technically correct.

It says that the reset policy is designed to achieve a distribution of final states that is equivalent to a starting distribution on the problem.  This is technically fairly difficult, as a problem, and I don't think it can be achieved through standard RL methods.   Later, it is clearer that there is a set of possible start states and they are all treated as goal states from the perspective of the reset policy.   That is a start set, not a distribution.  And, there's no particular reason to think that the reset policy will not, for example, always end up returning to a particular state.

Another point is that training a set of Q functions from different starting states generates some kind of an ensemble, but I don't think you can guarantee much about what sort of a distribution on values it will really represent.   Q learning + function approximation can go wrong in a variety of ways, and so some of these values might be really gross over or under estimates of what can be achieved even by the policies associated with those values.

A final, higher-level, methodological concern is that, it seems to me, as the domains become more complex, rather than trying to learn two (or more) policies, it might be more effective to take a model-based approach, learn one model, and do reasoning to decide how to return home (and even to select from a distribution of start states) and/or to decide if a step is likely to remove the robot from the "resettable" space.

All this aside, this seems like a fairly small but well considered and executed piece of work.  I'm rating it as marginally above threshold, but I indeed find it very close to the threshold.

---

> ### Author Response · Authors · 2017-12-12
> **Response to Reviewer1**
>
> We thank AnonReviewer1 for noting the main goal of our paper and recognizing how we incorporate safety using the learned reset policy, and further thank the reviewer for finding our paper a “well considered and executed piece of work.” We’ve addressed the concerns raised by the reviewer with clarifications in the main text, which we detail below.
>
> Assumption that environment is reversible - This assumption is indeed a limitation of our approach, which we note in the paper (Section 4 paragraph 2). We have expanded this section to clarify this detail. We show experimentally that our method can be applied to a number of realistic tasks, such as locomotion and manipulation. Extending our work to tasks with irreversible goal states is a great idea, and would make for interesting future work.
>
> Initial state distribution - You correctly note that the reset policy might always reset to the same state, thus failing to sample from the full initial state distribution. We have corrected this technical error in the revised version of the paper (Section 4, paragraph 2) by adding the additional assumption that the initial state distribution be unimodal and have narrow support. We also expanded the discussion of “safe sets” in Section 4.2 paragraph 1 to clarify the difference between the initial state distribution, the reset policy’s reward, and the safe set. We also describe a method to detect if there is mismatch between the the initial state distribution and the reset policy final state distribution.
>
> Q functions - We learn an ensemble of Q functions, each of which is a sampled from the posterior distribution over Q functions given the observed data. We expanded Section 4.4 paragraph 1 to note how this technique has been established in previous work (“Deep Exploration via Bootstrapped DQN” [Osband 2016] and “UCB Exploration via Q-Ensembles” [Chen 2017]).  In general, we are not guaranteed that samples from a distribution are close to its mean. However, our experiments on ensemble aggregation (taking the min, mean or max over the Q functions) had little effect on policy reward. If “gross under/over-estimation” had occurred, taking the min/max over the ensemble would have resulted in markedly lower reward. We expanded Appendix A paragraph 2 to explain this finding.
>
> Model-based alternative - We appreciate the comment regarding a potential model-based alternative to our method. However, we are not aware of any past model-based methods for solving this task. We would be happy to attempt a comparison or add a discussion if the reviewer has a particular prior method in mind. The early aborts in our method provide one method of identifying irreversible states. A model-based alternative could also serve this function. We believe that our early aborts, which only require learning a single reset policy, are simpler than learning a model of the environment dynamics and hypothesize that our approach will scale better to complex environments.

---

### Official Review · AnonReviewer3 · 2017-11-28
**Simple, practical approach for autonomous resets; good experimental results and ablation studies, but no real world tasks.**

**Rating:** 7
**Confidence:** 4

**Review:**

The paper solves the problem of how to do autonomous resets, which is an important problem in real world RL. The method is novel, the explanation is clear, and has good experimental results.

Pros:
1. The approach is simple, solves a task of practical importance, and performs well in the experiments.
2. The experimental section performs good ablation studies wrt fewer reset thresholds, reset attempts, use of ensembles.

Cons:
1. The method is evaluated only for 3 tasks, which are all in simulation, and on no real world tasks. Additional tasks could be useful, especially for qualitative analysis of the learned reset policies.
2. It seems that while the method does reduce hard resets, it would be more convincing if it can solve tasks which a model without a reset policy couldnt. Right now, the methods without the reset policy perform about equally well on final reward.
3. The method wont be applicable to RL environments where we will need to take multiple non-invertible actions to achieve the goal (an analogy would be multiple levels in a game). In such situations, one might want to use the reset policy to go back to intermediate “start” states from where we can continue again, rather than the original start state always.

Conclusion/Significance: The approach is a step in the right direction, and further refinements can make it a significant contribution to robotics work.

Revision: Thanks to the authors for addressing the issues I raised, I revise my review to 7

---

> ### Author Response · Authors · 2017-12-12
> **Response to Reviewer3**
>
> We thank AnonReviewer3 for recognizing the importance of the problem we aim to solve, and for noting that our simple method is supported with “good ablation studies.” We have addressed the issues raised by the reviewer, as discussed below:
>
> 1. We have run experiments on two additional environments (ball in cup and peg insertion), so the revised version of the paper shows experiments on 5 simulated environments (Section 6, paragraph 1). Videos of the learned policies visualize the learned forward + reset policies are available on the project website: https://sites.google.com/site/mlleavenotrace/
> Experimental evaluation of five distinct domains compares favorably to most RL papers that have appeared in ICLR in the past. While we agree that real-world evaluation of our method would be excellent, this is going substantially beyond the typical evaluation for ICLR RL work.
>
> 2. We ran additional environments that show that, in certain difficult situations, our method can solve tasks which a model without a reset policy cannot. Newly-added Sections 6.1 and 6.6 demonstrate this result in two settings. We summarize these results below in a separate post entitled “Additional Experiments.”
>
> 3. We expanded Section 4 paragraph 2 to clarify our assumption that there exists a reversible goal state. This is indeed a limitation of our approach, which we note in the paper (Section 4 paragraph 2). We show experimentally that our method can be applied to a number of realistic tasks, such as locomotion and manipulation. Extending our work to tasks with irreversible goal states by resetting to intermediate goals is a great idea, and would make for interesting future work.

---

### Official Review · AnonReviewer2 · 2017-12-16
**Interesting idea**

**Rating:** 5
**Confidence:** 4

**Review:**

(This delayed review is based on the deadline version of the paper.)

This paper proposes to learn by RL a reset policy at the same time that we learn the forward policy, and use the learned reset Q-function to predict and avoid actions that would prevent reset — an indication that they are "unsafe" in some sense.

This idea (both parts) is interesting and potentially very useful, particularly in physical domains where reset is expensive and exploration is risky. While I'm sure the community can benefit from ideas of this kind, it really needs clearer presentations of such ideas. I can appreciate the very intuitive and colloquial style of the paper, however the discussion of the core idea would benefit from some rigor and formal definitions.

Examples of intuitive language that could be hiding the necessary complexities of a more formal treatment:

1. In the penultimate paragraph of Section 1, actions are described as "reversible", while a stochastic environment may be lacking such a notion altogether (i.e. there's no clear inverse if state transitions are not deterministic functions).

2. It's not clear whether the authors suggest that the ability to reset is a good notion of safety, or just a proxy to such a notion. This should be made more explicit, making it clearer what this proxy misses: states where the learned reset policy fails (whether due to limited controllability or errors in the policy), that are nonetheless safe.

3. In the last paragraph of Section 3, a reset policy is defined as reaching p_0 from *any* state. This is a very strong requirement, which isn't even satisfiable in most domains, and indeed the reset policies learned in the rest of the paper don't satisfy it.

4. What are p_0 and r_r in the experiments? What is the relation between S_{reset} and p_0? Is there a discount factor?

5. In the first paragraph of Section 4.1, states are described as "irreversible" or "irrecoverable". Again, in a stochastic environment a more nuanced notion is needed, as there may be policies that take a long time to reset from some states, but do so eventually.

6. A definition of a "hard" reset would make the paper clearer.

7. After (1), states are described as "allowed". Again, preventing actions that are likely to hinder reset cannot completely prevent any given state in a stochastic environment. It also seems that (2) describes states where some allowed action can be taken, rather than states reachable by some allowed action. For both reasons, Algorithm 1 does not prevent reaching states outside S*, so what is the point of that definition?

8. The paper is not explicit about the learning dynamics of the reset policy. It should include a figure showing the learning curve of this policy (or some other visualization), and explain how the reset policy can ever gain experience and learn to reset from states that it initially avoids as unsafe.

9. Algorithm 1 is unclear on how a failed reset is identified, and what happens in such case — do we run another forward episode? Another reset episode?

---

> ### Author Response · Authors · 2017-12-24
> **Response to Reviewer2**
>
> Thank you for the comments! It seems that all the concerns have to do with the writing in the paper and are straightforward to fix. We have addressed all the concerns raised about the paper in this review. Given that all issues have been addressed, we would appreciate if the reviewer could take another look at the paper.
>
> 1. We have clarified our definition of reversible action in Section 1 paragraph 4. For deterministic MDPs, we say an action is reversible if it leads to a state from which there exists a reset policy that can return to a state with high density under the initial state distribution. For stochastic MDPs, we say an action is reversible if the probability that an oracle reset policy that can reset from the next state is greater than some safety threshold. Note that definition for deterministic MDPs is a special case of the definition for stochastic MDPs.
>
> 2. The ability of an oracle reset policy to reset is a good notion of safety. In our algorithm, we approximate this notion of safety, assuming that whether our learned reset policy can reset in N episodes is a good proxy for whether an oracle reset policy can reset. We have clarified Section 1 paragraph 4 to make this distinction clear. We also added Appendix B to discuss handling errors in Q value estimation. In this section, we describe how Leave No Trace copes with overestimates and underestimates of Q values.
>
> 3. We have corrected this technical error in Section 3 paragraph 2 by redefining the reset policy as being able to reach p_0 from any state reached by the forward policy. That our learned reset policy only learns to reset from states reached by the forward policy is indeed a limitation of our method. However, note that early aborts help the forward policy avoid visiting states from which the reset policy is unable to reach p_0.
>
> 4. For the continuous control environments, the initial state distribution p_0 is uniform distribution centered at a “start pose.”  We use a discount factor \gamma = 0.99. Both details have been noted in Appendix F.3 paragraph 2. The reset reward r_r is a hand-crafted approximation to p_0. For example, in the Ball in Cup environment, r_r is proportional to the negative L2 distance from the ball to the origin (below the cup). For cliff cheetah, r_r includes one term that is proportional to the distance of the cheetah to the origin, and another term indicating whether the cheetah is standing. S_{reset} is the set of states where r_r(s) is greater than 0.7 (Appendix C.3 paragraph 2)
>
> 5. We have clarified Section 4.1 paragraph 1 to explain how our proposed algorithm handles both cases: states from which it is impossible to reset and states from which resetting would take prohibitively many steps. In both cases, the cumulative discounted reward (and hence the value function) will be low. By performing an early abort when the value function is low, we avoid both cases.
>
> 6. We added a definition of “hard reset” to Section 4.2 paragraph 1: A hard reset is an action that resamples that state from the initial state distribution. Hard resets are available to an external agent (e.g. a human) but not the learned agent.
>
> 7. We acknowledge that the proposed algorithm does not guarantee that we never visit unsafe states. In Appendix A, we have added a proof that Leave No Trace would only visit states that are safe in expectation if it had access to the true Q values. Appendix A.3 discusses the approximations we make in practice that can cause Leave No Trace to visit unsafe states. Finally, Appendix B discusses how Leave No Trace handles errors incurred by over/under-estimates of Q values.
>
> 8. Newly added Appendix D visualizes the training dynamics by plotting the number of time steps in each episode before an early abort. Initially, early aborts occur near the initial state distribution, so the forward episode lengths are quite short. As the reset policy improves, early aborts occur further from the initial state, as indicated by longer forward episode lengths. Newly added Appendix B discusses how Leave No Trace handles errors incurred by over/under-estimates of Q values. It describes how Leave No Trace learns that an “unsafe” state is actually safe.
>
> 9. We detect failed resets in line 12 of Algorithm 1. We have added a comment to help clarify this. When a failed reset is detected, a hard reset occurs (line 13).

---

### Official Review · AnonReviewer4 · 2018-01-04
**Great idea but the write up needs to be made clearer**

**Rating:** 7
**Confidence:** 4

**Review:**

This paper proposes the idea of having an agent learning a policy that resets the agent's state to one of the states drawn from the distribution of starting states. The agent learns such policy while also learning how to solve the actual task. This approach generates more autonomous agents that require fewer human interventions in the learning process. This is a very elegant and general idea, where the value function learned in the reset task also encodes some measure of safety in the environment.

All that being said, I gave this paper a score of 6 because two aspects that seem fundamental to me are not clear in the paper. If clarified, I'd happily increase my score.

1) *Defining state visitation/equality in the function approximation setting:* The main idea behind the proposed algorithm is to ensure that "when the reset policy is executed from any state, the distribution over final states matches the initial state distribution p_0". This is formally described, for example, in line 13 of Algorithm 1.
The authors "define a set of safe states S_{reset} \subseteq S, and say that we are in an irreversible state if the set of states visited by the reset policy over the past N episodes is disjoint from S_{reset}." However, it is not clear to me how one can uniquely identify a state in the function approximation case. Obviously, it is straightforward to apply such definition in the tabular case, where counting state visitation is easy. However, how do we count state visitation in continuous domains? Did the authors manually define the range of each joint/torque/angle that characterizes the start state? In a control task from pixels, for example, would the exact configuration of pixels seen at the beginning be the start state? Defining state visitation in the function approximation setting is not trivial and it seems to me the authors just glossed over it, despite being essential to your work.

2) *Experimental design for Figure 5*: This setup is not clear to me at all and in fact, my first reaction is to say it is wrong. An episodic task is generally defined as: the agent starts in a state drawn from the distribution of starting states and at the moment it reaches the goal state, the task is reset and the agent starts again. It doesn't seem to be what the authors did, is that right? The sentence: "our method learns to solve this task by automatically resetting the environment after each episode, so the forward policy can practice catching the ball when initialized below the cup" is confusion. When is the task reset to the "status quo" approach? Also, let's say an agent takes 50 time steps to reach the goal and then it decides to do a soft-reset. Are the time steps it is spending on its soft-reset being taken into account when generating the reported results?


Some other minor points are:

- The authors should standardize their use of citations in the paper. Sometimes there are way too many parentheses in a reference. For example: "manual resets are necessary when the robot or environment breaks (e.g. Gandhi et al. (2017))", or "Our methods can also be used directly with any other Q-learning methods ((Watkins & Dayan, 1992; Mnih et al., 2013; Gu et al., 2017; Amos et al., 2016; Metz et al., 2017))"

- There is a whole line of work in safe RL that is not acknowledged in the related work section. Representative papers are:
    [1] Philip S. Thomas, Georgios Theocharous, Mohammad Ghavamzadeh: High-Confidence Off-Policy Evaluation. AAAI 2015: 3000-3006
    [2] Philip S. Thomas, Georgios Theocharous, Mohammad Ghavamzadeh: High Confidence Policy Improvement. ICML 2015: 2380-2388

- In the Preliminaries Section the next state is said to be drawn from s_{t+1} ~ P(s'| s, a). However, this hides the fact the next state is dependent on the environment dynamics and on the policy being followed. I think it would be clearer if written: s_{t+1} ~ P(s'| s, \pi(a|s)).

- It seems to me that, in Algorithm 1, the name 'Act' is misleading. Shouldn't it be 'ChooseAction' or 'EpsilonGreedy'? If I understand correctly, the function 'Act' just returns the action to be executed, while the function 'Step' is the one that actually executes the action.

- It is absolutely essential to depict the confidence intervals in the plots in Figure 3. Ideally we should have confidence intervals in all the plots in the paper.

---

> ### Author Response · Authors · 2018-01-04
> **Response to Reviewer 4**
>
> We thank Reviewer 4 for the comments and for finding our paper a “very elegant and general idea.” The main comments had to do with clarity - we have addressed these in the revised version. We would appreciate of the reviewer would reevaluate the paper given the new clarifications.
>
> 1. (State visitation) We implement our algorithm in a way that avoids having to test whether two states are equal (indeed, a challenging problem). In Equation 4, we define the set of safe states S_{reset} implicitly using the reset reward function r_r(s). In particular, we say a state is safe if the the reset reward is greater than some threshold (0.7 in our experiments). For example, in the pusher task, the reset reward is the distance from a certain (x, y, z) point, so S_{reset} is the set of points within some distance of this point. We added a comment to line 13 of the algorithm clarifying how we test if a state is in S_{reset}.
>
> 2. (Figure 5) We clarified our description of the environments in Section 6 paragraph 1 to note that the episode is not terminated when the agent reaches a goal state. For the experiment in Figure 5, the ‘forward-only’ baseline and our method are non-episodic - the environment is never reset and no hard resets are used (Section 6.1). The ‘status quo’ baseline is episodic, doing a hard reset every T time steps (for ball in cup, T = 200 steps). We reworded the confusing sentence about our method as follows: “In contrast, our method learns to solve this task by automatically resetting the environment after each attempt, so the forward policy can practice catching the ball without hard resets.” All results in the paper include time steps for both the forward task and the reset task. We clarified this in Section 6.1 paragraph 1. This highlights a strength of our method: even though the agent spends a considerable amount of time learning the reset task, it still learns to do the forward task in roughly the same number of total steps (steps for forward task + steps for reset task).
>
>
> Minor points:
>
> 1. Citations. We’ve removed the extra parenthesis for the Q-learning references. Generally, we use “e.g.” in citations when the citation is an example of the described behavior.
>
> 2. Thanks for the additional references! We’ve included them in Section 2 paragraph 1.
>
> 3. We chose to separate out the policy from the transition dynamics. Action a_{t} is sampled from \pi(a_{t} | s_{t}) and depends on the policy; next state s_{t} is sampled from P(s_{t+1} | s_{t}, a_{t}) and depends on the transition dynamics.
>
> 4. Good idea. We’ve  changed “Act()” to “ChooseAction()” in Algorithm 1.
>
> 5. For Figure 3, we agree confidence intervals would be helpful. We can’t regenerate the plot in the next 24 hours before the rebuttal deadline, but will include confidence intervals in the camera-ready version.

---

> > ### Comment · AnonReviewer4 · 2018-01-04
> > **Figure 5 is still not clear for me**
> >
> > 1. (State visitation) Thanks for clarifying that. However, it seems to me then that each domain needs to have an $r_r$ reward function hand-engineered to describe the proximity to the start state, right? In the authors' example, "in locomotion experiments, the reset reward is large when the agent is standing upright", what "standing upright" means? Did the authors had to specify a specific joint configuration? Did they use the initial state as joint configuration? Even if they didn't, I assume a distance/divergence metric had to be defined when generating $r_r$, right? Ideally I'd like to see that described in the main paper, so we can judge how hard it is to obtain $r_r$.
> >
> > 2. (Figure 5) I'm sorry, the clarification didn't help me. When the authors say: "Once the forward-only approach catches the ball, it gets maximum reward by keeping the ball in the cup", does it mean that the agent receives $R_max$ reward at each time step after the ball is in the cup? Let's say episodes are 100 time steps long, and that achieving the goal leads to a reward of +1. If "forward-only" approach reaches the goal in the time step 40, what is going to be the return of that episode? Will it be +1? What if the reset approach does the same thing, will it get a +1 in the time step 40, reset, and get another +1 in time step, let's say, 80? I hope I was able to clarify my question. It seems to me that the number of steps each agent is using is being counted in an unfair way.
> >
> > Minor points: Thanks, for addressing those, I think the paper is clearer after these changes.

---

> > > ### Author Response · Authors · 2018-01-05
> > > **Clarifying Figure 5**
> > >
> > > 1. You correctly note that a reset reward function r_r(s) must be specified for each task. In practice, we found that a very simple design worked well for our experiments. We used the negative distance to some start state, plus any reward shaping included in the forward reward. We clarified Section 4 paragraph 1 to note this.
> > >
> > >
> > > 2. To ensure fair evaluation of all approaches, we used a different procedure for evaluation than for training. For all the figures showing reward in the paper (including Fig 5), we evaluate the performance of a policy at a checkpoint by creating a copy of the policy in a separate thread, running only the forward policy (not the reset policy) for a fixed number of steps T and computing the average per-step reward (cumulative reward divided by T). We added these details to Section 6 paragraph 1.
> > >
> > > For example, if the “forward-only” approaches catches the ball at t=40 and keeps the ball in the cup through t=100, then the average per-step reward is (100 - 40) / 100 = 0.6. For our approach, we only run the forward policy during evaluation. It it catches the ball at t=40 and keeps the ball in the cup through t=100, its average per-step reward is also 0.6.

---

> > > > ### Comment · AnonReviewer4 · 2018-01-05
> > > > **It seems to me that your comparison is unfair.**
> > > >
> > > > 1. So, did you use the Euclidean distance between two states using what coordinates? The absolute coordinates (x, y, z, and speeds, for example), between the initial state and the current state? Or did you do so through the representation learned by the network? That is not clear, what makes these results irreproducible. Can you add a paragraph after each section explicitly describing what you used as start state/representation/distance metric? Otherwise I can't judge it because I wouldn't be able to reproduce the obtained results.
> > > >
> > > > 2. But following this procedure, how can you disentangle the benefit of simply training the agent for more time steps from the fact that you are actually resetting the policy? Let's say that the agent can reliably solve the task in 50 episodes. One of the agents (the one with the reset policy) would get twice as much data when learning how to solve the task. It seems an unfair comparison that generates misleading results. As far as I understand, it could have nothing to do with the task itself, but with the amount of data being used. This suggests a much deeper discussion in fact, that is related to how to evaluate agents. You are comparing the performance of an agent evaluated at training time with an agent that has a test time. That's unfair, isn't it?

---

> > > > > ### Author Response · Authors · 2018-01-05
> > > > > **Clarifying Fair Comparison**
> > > > >
> > > > > 1. We directly use Euclidean distance between observations. The start state was chosen by sampling from the initial state distribution once. We use the same start state for all experiments with a given environment. We have added these details to Appendix F.2
> > > > >
> > > > >
> > > > > 2. The agent trained with the reset policy does *not* get twice as much data -- it gets the same number of transitions, and has to spend these transitions on both the reset and forward policy.
> > > > >
> > > > > All methods compared observe the same amount of data (Section 6 paragraph 1). For example, in Figure 5, all approaches observe 1 million steps. While the “status quo” baseline and the “forward-only” baseline spend all 1 million steps trying to catch the ball, our approach spends a total of 0.5 million steps trying the catch the ball, and 0.5 million steps trying to reset the ball. Our experiments show that given equal amounts of data and using identical evaluation procedures (Section 6 paragraph 1), our method performs significantly fewer manual resets, while still learning to complete the task.

---

> > > > > > ### Comment · AnonReviewer4 · 2018-01-06
> > > > > > **Thanks for the clarifications**
> > > > > >
> > > > > > Thanks for these clarifications. The new section in the Appendix clarifies things a little bit more (although I'd recommend the authors to get rid of sentences such as 'certain dimensions' and to be more specific about it).
> > > > > >
> > > > > > I'll increase my score (from 6 to 7) after our discussion. You have ensured me that the agent is using the exact same number of transitions in both settings in Figure 5. I still think the paragraph is poorly written, it is confusing. Sentences such as:  Once the forward-only approach catches the ball, it gets maximum reward by keeping the ball in the cup" still throw me off. The episode should be reset and that's it (hard reset).  But the authors have ensured me this comparison is fair.
> > > > > >
> > > > > > I'll not drastically increase my score because I think the writing could be improved substantially to make the paper clearer. Also, the need to define a distance metric and a start state is not as general as I'd like it to be. For example, in vision-based tasks, would the difference be measured as distance between all pixels? It seems wrong. Naturally, I don't expect the authors to address all these questions, but since the method is not so general, I'm not thrilled about it. All that being said, I think the paper has an interesting idea so it should be accepted.

---

### Author Response · Authors · 2017-12-12
**Additional experiments in revised version**

We ran two additional experiments for the revised version of the paper. These experiments motivate learning a reset controller in two settings: when hard resets are available and when hard resets are not available.

First, we consider the setting where hard resets are not available (Section 6.1). We use an environment (“ball in cup”) where the agent controls a cup attached to a ball with a string, and attempts to swing the ball up into the cup. The agent receives a reward of 1 if the ball is in the cup, and 0 otherwise. We don’t terminate the episode when the ball is caught, so the agent receives a larger reward the longer it keeps the ball in the cup. Using Leave No Trace, the agent repeatedly catches the ball and then pops the ball out of the cup to reset. In contrast, we compare to a baseline (“forward only”) that simply maximizes its environment reward. Once this baseline catches the ball once during random exploration, it trivially maximizes its reward by doing nothing so the ball stays in the cup. However, this baseline has failed to learn to catch the ball when the ball is initialized below the cup. The video below illustrates the training dynamics of the baseline (left) and Leave No Trace (right): https://www.youtube.com/watch?v=yDcFMR59clI

Second, we consider the setting where hard resets are allowed (but ideally avoided) (Section 6.6). We use the peg insertion task, where the agent controls an arm with a peg at the end. The agent receives a reward of 1 when the peg is in the hole and 0 otherwise (plus a control penalty). Learning this task is very challenging because the agent receives no reward during exploration to guide the peg towards the hole. The baseline (“status quo”) starts each episode with the peg far from the hole, and is hard reset to this pose after each episode. Our approach uses Leave No Trace to solve this task in reverse, so the peg insertion task corresponds to the reset policy. (We compare to reverse curriculum learning [Florensa 2017]) in Section 2 paragraph 3.) We start each episode with the peg inside the hole. The forward policy learns to remove the peg until an early abort occurs, at which point the reset policy inserts the peg back in the hole. As learning progresses, early aborts occur further and further from the hole. During evaluation, we initialize both the baseline and Leave No Trace with the peg far from the hole. Our method successfully inserts the peg while the baseline fails. Importantly, this result indicates that learning a reset policy enables our method to learn tasks it would otherwise be unable to solve, even when avoiding hard resets is not the primary goal.

---

### Decision · Program_Chairs · 2018-01-29
**ICLR 2018 Conference Acceptance Decision**

**Decision:**

Accept (Poster)

**Comment:**

This paper is an easy accept -- three reviewers have above threshold scores, while one reviewer is slightly below threshold, but based on the submitted manuscript.  It appears that the paper has substantially improved based on reviewer comments.

Pros:

All reviews had positive sentiment: "very elegant and general idea" (Reviewer4); "idea is interesting and potentially very useful" (Reviewer2); "method is novel, the explanation is clear, and has good experimental results" (Reviewer3); "a good way to learn a policy for resetting while learning a policy for solving the problem.  Seems like a fairly small but well considered and executed piece of work." (Reviewer1)

Cons:

One reviewer found that testing in only three artificial tasks was a limitation.

The initial reviews noted several issues where clarification of the text and/or figures was needed.  There were also a bunch of statements where the reviewers questioned the technical correctness / accuracy of the discussion.  Most of these points appear to have been adequately addressed in the revised manuscript.